# Donor Conditioning and Organ Pre-Treatment Prior to Kidney Transplantation: Reappraisal of the Available Clinical Evidence

**DOI:** 10.3390/jcm13144073

**Published:** 2024-07-12

**Authors:** Peter Schnuelle, Bernhard K. Krämer

**Affiliations:** 1Center for Renal Diseases, Academic Teaching Practice of the University Medical Center Mannheim, University of Heidelberg, 69469 Weinheim, Germany; 2Vth Department of Medicine, University Medical Center Mannheim, University of Heidelberg, 68167 Mannheim, Germany; bernhard.kraemer@umm.de

**Keywords:** cold ischaemia, delayed graft function, donor conditioning, dopamine, graft survival, hypothermic machine perfusion, kidney transplantation, organ preservation, primary nonfunction, randomised controlled trial, therapeutic hypothermia

## Abstract

Therapeutic measures aimed at optimising organ function prior to transplantation—whether by conditioning the donor after determination of brain death or by improving organ preservation after kidney removal—have the potential to enhance outcomes after transplantation. The particular advantage is that, unlike any optimised immunosuppressive therapy, a favourable effect can be achieved without side effects for the organ recipient. In recent years, several such measures have been tested in controlled clinical trials on large patient cohorts following kidney transplantation. Hypothermic pulsatile machine perfusion, in particular, has become the focus of interest, but interventions in the donor prior to organ removal, such as the administration of low-dose dopamine until the start of cold perfusion as an example of conditioning antioxidant therapy and therapeutic donor hypothermia in the intensive care unit after brain death confirmation, have also significantly reduced the frequency of dialysis after transplantation with far less effort and cost. With regard to benefits for graft survival, the database for all procedures is less clear and controversial. The aim of this review article is to re-evaluate the available clinical evidence from large multicentre controlled trials, which have also significantly influenced later meta-analyses, and to assess the significance for use in routine clinical practice.

## 1. Introduction

Transplantation is the best possible treatment for patients with end-stage renal disease. However, there is a large discrepancy worldwide between the need and availability of organs for transplantation. Considerable efforts have therefore been made to make optimal use of the limited resource of transplantable kidneys through sophisticated allocation algorithms and improved logistics to shorten ischaemia times. Advances in immunosuppressive therapy have steadily increased success rates after kidney transplantation [1,2]. The quality of donor kidneys can also be improved with therapeutic intervention prior to transplantation [3]. This includes organ-protective measures that start already in the intensive care unit (ICU) after confirmation of brain death, including the administration of conditioning drugs and the use of machine perfusion (MP) after the removal of the kidneys from the donor organism. In recent years, a number of large controlled clinical trials (RCT) on this topic have been published in highly ranked journals. While the administration of steroid boluses to reduce post-ischaemic renal transplant failure or intravenous levothyroxine for the stabilisation of cardiac function failed to demonstrate any effectiveness [4,5], therapeutic donor hypothermia [6], donor pre-treatment with low-dose dopamine [7], and hypothermic pulsatile MP [8,9] each reduced the need for dialysis after transplantation. However, the effects on long-term graft survival are less clear. An improvement in initial graft function is undoubtedly a success. Nevertheless, a detailed examination of the data leaves several questions unanswered that are related to the ambivalence of the study endpoint—delayed graft function (DGF)—and must be discussed in terms of cost and benefit. The aim of this article is to take another critical look at the available evidence.

## 2. Post-Operative Dialysis as a Clinical Endpoint

The requirement of dialysis immediately after transplantation was investigated in all the controlled clinical trials discussed in this article. In the hypothermia and MP studies, only one post-operative dialysis in the first week after transplantation was taken to assess efficacy according to the internationally recognised definition of DGF, while repeated post-operative dialysis until the onset of graft function was considered the primary endpoint in the dopamine study. The advantage of the latter definition is that it is less susceptible to an indication bias. To date, there are no generally applicable indication criteria for dialysis after transplantation. Its indication is often based solely on the post-operative condition of the recipients, including their laboratory values, and is subject to the subjective assessment of the treating nephrologist. A single dialysis does not necessarily mean an initially impaired graft function. For example, a single dialysis may only be required to correct an early post-operative fluid or electrolyte imbalance (hyperkalaemia). Repeated post-operative dialysis as an endpoint, on the other hand, more accurately reflects graft dysfunction. The duration of post-operative dialysis dependency has been shown to correlate with an unfavourable prognosis for the graft [10], while a single dialysis leads to comparable long-term results in kidneys that have resumed function without post-operative dialysis [11].

## 3. Therapeutic Donor Hypothermia

In 2015, the results of a large multicentre intervention study of therapeutic donor hypothermia were published [6]. Circulatory stable organ donors after determination of death, according to neurological criteria, were randomly assigned to one of two targeted temperature ranges: 34 to 35 °C (hypothermia) or 36.5 to 37.5 °C (normothermia). The initiation and maintenance of therapeutic hypothermia in the ICU took place over a period of 16 to 24 h after the determination of brain death. Four adverse events occurred in the organ donors: one episode of cardiac arrhythmia and one episode of systemic hypertension in the hypothermia group, and two episodes of cardiac arrest prior to organ removal in the normothermia group. Since the average time between the determination of brain death and the start of cold perfusion in the USA is about 24 h [12], no additional costs were incurred. The study intervention was conducted in two organ donor regions in the USA and investigated the incidence of DGF in the recipient centres where the transplantation was performed. The trial was terminated prematurely after an interim analysis showed highly significant efficacy after the enrolment of 370 of the planned 500 donors. At least one kidney was transplanted from 302 of these donors, and 566 recipients had complete outcome data. DGF occurred in 79 of 280 (28.2%) transplant recipients from the hypothermia group and in 112 of 286 (39.2%) from the normothermia group (*p* = 0.008). The beneficial effect of induced hypothermia was particularly pronounced in the subgroup of extended criteria donors (ECDs) with a DGF rate of 31.0% compared to 56.5% in the control group (*p* = 0.003).

In retrospect, the favourable effect of an organ donor with a lower core body temperature on initial graft function was also observed in the database of the randomised dopamine trial. However, there was no benefit for long-term graft survival [13]. This was later confirmed by the investigators of the hypothermia study, who overall failed to demonstrate a significant graft survival benefit 1 year after transplantation (*p* = 0.15). Surprisingly, there was a small survival advantage of 4% in kidney transplants from standard criteria donors (SCDs) only, which just reached the significance level [14]. However, this finding was put into perspective in a more recent RCT conducted by the same group [15]. This trial involved a total of 934 kidney transplants exclusively from SCDs. There was no difference in the post-operative dialysis frequency: 17% in the hypothermia group vs. 18% in the normothermia group, with an adjusted odds ratio (OR) for DGF of 0.92 (95% confidence interval [CI] 0.64–1.33, *p* = 0.66).

## 4. Dopamine

Why donor dopamine? As early as the late 1990s, retrospective observational clinical studies from our own centre indicated that kidney transplant recipients from a donor who had received dopamine in the ICU prior to organ removal had better early function after transplantation and required less post-operative dialysis [16,17]. At that time, the administration of low-dose dopamine to supposedly stabilise renal function in the ICU was still widespread. Controlled clinical data have now clearly demonstrated that dopamine is not able to prevent or shorten acute renal failure in critically ill patients [18]. On the contrary, as outlined in a 2003 review article by Holmes and Walley entitled “Bad medicine: low-dose dopamine in the ICU”, even renal doses of dopamine can cause unfavourable side effects, worsen splanchnic oxygenation, impair the gastrointestinal tract, negatively affect the endocrine and immunological systems, and weaken respiratory drive [19]. All of these side effects are mediated via adrenergic or dopaminergic receptors. For this reason, dopamine has now been largely eliminated from intensive-care therapy.

The data situation described above has certainly made the general acceptance of donor dopamine for conditioning the kidneys in the ICU prior to transplantation more difficult, but it does not argue against its efficacy. Due to a pleiotropic mechanism of action, protection is mediated via dopamine’s antioxidant property rather than a receptor effect. Under the conditions of cold preservation, oxidative stress, among other things, occurs through the degradation of haemproteins (cytochrome P450) with the resulting release of catalytic iron ions [20,21]. Oxidative stress, in turn, leads to a release of calcium from intracellular stores and an increased influx of calcium from the extracellular space [22,23]. Calcium must be pumped out of the cell interior to maintain cellular homeostasis. This consumes energy-rich phosphates that cannot be regenerated under cold conditions. Calcium accumulation inside the cell also leads to damage to the mitochondrial membrane with the consequence of an increased leakage of oxygen radicals into the cytosol. A vicious circle is set in motion until the mitochondrial membrane potential finally collapses [24]. It has been shown that dopamine slows down the vicious circle of intracellular calcium accumulation and ATP loss due to its reducing properties and can thus delay cold ischaemia damage. The prerequisite is that dopamine has been accumulated in sufficient concentration in the intracellular space prior to cold preservation [25].

Dopamine and all-natural and synthetic catecholamines have a benzene ring hydroxylated in the 3,4 ortho position. As a result, the molecule has reducing properties and can absorb free electrons from oxygen radicals, which are increasingly produced under cold preservation conditions. Conformational change by dihydroxylation in the 3,5 meta position, on the other hand, leads to a loss of the reducing effect, which also completely abrogates protection against cold preservation injury. The N-terminal residue, which distinguishes the various vasoactive derivatives from each other, determines their affinity to the adrenergic or dopaminergic receptors. The N-acylation of dopamine increases lipophilicity and significantly improves the efficacy of protection. Derivatives such as N-octanoyl-dopamine (NOD) would be advantageous for the conditioning of donors because they no longer have a haemodynamic effect [26]. However, they are not authorised in humans. Dopamine, therefore, represents a compromise to a certain extent, as dopamine has the fewest vasoactive side effects compared to other catecholamines at a given equivalent dose. For these side effects, the brain-dead donor needs to be closely monitored in the ICU to prevent circulatory destabilisation.

In a prospective randomised clinical trial, organ donors received either a standard dopamine infusion of 4 µg/kg·min until cross-clamping or no infusion after the confirmation of brain death. Donors were included if they were circulatory stable with only a minimal dose of noradrenaline (<0.4 µg/kg·min) and had a serum creatinine < 1.3 mg/dL on admission to the ICU. Kidneys were allocated to recipients centrally by Eurotransplant on the basis of waiting time and HLA matching. The primary endpoint was the need for more than one dialysis after transplantation. The study evaluation was conducted in 60 European transplant centres. Donor dopamine significantly reduced the need for repeated dialyses (24.7 vs. 35.4%, *p* = 0.01). A post hoc analysis revealed that the effect of donor organ conditioning was quantitatively greatest in the subgroup with the longest cold ischaemia (>17 h), which fits well with the underlying molecular mechanism of action. In addition, renal function and freedom from dialysis one week after transplantation correlated with the duration of the dopamine infusion [7]. Both the primary endpoint and the 5-year graft survival rate showed some saturation kinetics after an application time of slightly more than 7 h [27]—consistent with the aforementioned experimental findings that dopamine’s protection depends on its diffusion into the cell interior [25], which, in turn, is a time-dependent process. While no statistical effect on long-term graft survival could be demonstrated in the intention-to-treat analysis, there was a significant survival benefit when dopamine was administered for longer than 7 h (90.3% vs. 80.2%, log rank *p* = 0.04, after censoring for death with functioning graft). Due to circulatory side effects, the dopamine infusion had to be terminated prematurely in 15% of the donors. However, none of the donors were destabilised, and the function of their kidneys after transplantation was absolutely comparable to that of the control group that had not received a dopamine infusion.

## 5. Hypothermic Pulsatile Machine Perfusion

The European multicentre study on hypothermic pulsatile MP, published in N. Engl. J. Med. in 2009 [8], has given the method a worldwide boost, although the data situation has remained ambiguous due to non-negligible shortcomings in the conduct and analysis of the study. The primary endpoint, a reduction in DGF from 26.5 to 20.8%, was only borderline significant (*p* = 0.05) when one-sided statistical tests were used in the analysis. This was justified by the design of a formal paired study, in which the kidneys from one donor each were randomly assigned to MP or static cold storage. However, the paired design and, thus, the approach of a paired statistical analysis can also be questioned, as a donor’s kidneys often do not perform equally. In addition, the vascular anatomy (aberrant vessels), including the quality of surgical organ harvesting, often differs significantly between the right and left sides. More importantly, the efficacy of the study intervention was determined in two different recipients with individually different risks of requiring dialysis after transplantation, and, as mentioned above, this does not necessarily reflect graft function. In fact, no difference in creatinine clearance was observed at 14 days (42 vs. 40 mL/min, *p* = 0.25). One point really worthy of criticism is that the preservation method was changed in 25 donors (4.6%) for technical reasons (aortic patch too small/too many renal arteries for connection to the MP device) contrary to randomisation. Taking into account the relatively small study effect (reduction in post-operative dialysis incidence by 5.7%), this led to a non-negligible distortion in the allocation of kidneys to treatment. Assuming that a comparable number of kidneys in the control arm also did not allow a vascular connection to the MP device, the above-mentioned protocol violation resulted in approximately 9% of the statically cooled kidneys having a more complicated vascular anatomy than those in the MP arm. The accumulation of kidneys with more complicated vascular anatomy likely also increased the complexity of the surgical vascular anastomoses in the control arm and may have biassed the study results in favour of MP. As is well known, despite the attempt to adjust for the number of renal arteries, a bias cannot be eliminated by statistical adjustments in the data analysis. It is noteworthy that in the control arm, primary non-function (analysed as a secondary endpoint) was recorded in 4.8% of cases, while the rate in the machine-perfused kidneys was 2.1%. The difference was not considered significant in the original publication, with a *p*-value of 0.08. However, in contrast to the statistical analysis of the primary endpoint, this time, an unpaired test was applied. With a paired Fisher’s exact test, the *p*-value of 0.04 is even below the significance level determined for the primary outcome. This would make the European multicentre study the only randomised trial to date that has demonstrated a clinical effect of MP on initial graft failure. A later meta-analysis on this issue, including the European multicentre study, was unable to show any influence on primary non-function [28]. In this context, it should be mentioned that about one-third of early graft losses after kidney transplantation are due to technical failure or problems with vascular anastomoses, as large-registry data from the Netherlands and the UK consistently show [29]. Another notable omission is that seven donor pairs, in whom technical failure of the MP occurred after randomisation, as shown in the study flowchart in the original publication, were excluded from the analysis. The researchers did not specify whether these kidneys were still transplantable or, if transplantable, whether they failed or had initial or delayed graft function. In the event that the failure of MP was, in fact, a major adverse event of the trial intervention, a data analysis according to the intention-to-treat principle would be appropriate. If one additional kidney assigned to MP had failed, the primary trial outcome would have also missed the statistical significance level in the paired analysis.

The investigators of the European multicentre study reported a favourable effect of MP on the 1-year function rate of the transplanted kidneys (94 vs. 90%, log rank *p* = 0.04). The quantitative difference of 4% (91 vs. 87%) was also statistically significant when analysing the 3-year function rates [30]. After 1 year, the survival curves were parallel without further divergence. Thus, in purely mathematical terms, 67.5% of the survival advantage after 3 years can be attributed to the difference in initial non-function. Two meta-analyses, including the largest European multicentre study at the time, were unable to demonstrate a significant effect on graft survival [28,31]. Another meta-analysis, which only considered studies published after 2010, found no effect at all [32]. Nevertheless, in their conclusion, the authors of the Cochrane meta-analysis [28] made the following statement: “…There is strong evidence that hypothermic MP has a positive impact on transplant survival in both the short and long term, in both DBD and DCD grafts. This is to be expected given previous research has shown the DGF is associated with higher rates of kidney loss…” There may be subgroups that particularly benefit from MP, e.g., kidneys from donors after cardiac death or from brain-dead ECDs. In a separate analysis embedded in the European multicentre study, the focus was specifically on kidneys from the latter donor category [33]. There was a comparable reduction in DGF of 7.7% from 29.7 to 22%. At the same time, the incidence of initial non-function was four-fold higher in the controls (12 vs. 3%, *p* = 0.04). This resulted in a 1-year graft survival advantage in favour of MP of 92.3 vs. 80.2%, *p* = 0.02. However, in 5.5% of the ECDs, the perfusion method had also been switched contrary to the initial randomisation. The researchers’ conclusion that recipients of ECD kidneys particularly benefited from MP therefore remains limited against the background of the selection bias inherent in the study already discussed.

The bold prediction by Tingle et al. [28] cited above has since been relativised by a more recent American trial from 2023 [9]. This 3-arm RCT, the largest in number to date, aimed at comparing MP with or without intended donor hypothermia versus hypothermia alone and found no difference in 1-year graft survival. The Kaplan–Meier survival curves were exactly superimposed in all three groups, although in the two MP groups, the DGF rate was significantly reduced by 8 and 11%, respectively—with a comparable dialysis frequency in the control group, as in the European study, in which the body temperature was not lowered before organ removal. The study was originally intended to show that donor hypothermia is not inferior to MP. Therefore, the result—compared to the European multicentre study—was not to be expected due to the even greater reduction in the primary endpoint by MP. In the recent study, the treatment of the kidneys was also not always carried out in accordance with the randomisation. However, unlike in the European multicentre study, protocol violations contrary to randomisation, which occurred in 27% of the kidneys allocated to MP for technical or organisational reasons, were handled in the analysis according to the intention-to-treat principle, so that selection bias was avoided. It should be emphasised that the control group in the current study was not affected by an increased incidence of primary non-function.

The trial by Malinoski et al. [9] was designed and conducted as an open study. After assignment to the preservation method, no other specifications were made at the transplant centre level with regard to the organisational procedure and timing of transplantation. Therefore, two important questions arise, namely, whether the selected study endpoint DGF was able to specifically differentiate early graft dysfunction and whether the complex MP logistics justify the routine application of MP in light of the available findings. Both questions are naturally closely linked [34]. It is noticeable that the machine-perfused kidneys were only transplanted after a longer duration of cold ischaemia of 2.5 h on average [9]. Due to the large number of transplants performed in each study arm, it can hardly be assumed that the difference occurred by chance. Rather, in the recent past, the view prevailed that kidneys at the MP tolerate longer cold ischaemia, so that organisational processes in the transplant centres could be improved or even nocturnal transplants could be avoided [32]. Apart from simpler logistics in the transplant centres, allowing longer cold ischaemia times would also facilitate the improvement of HLA compatibility. A better HLA match influences long-term survival beyond 3 years of observation [35,36]. Longer follow-up times are not realistic in prospective RCTs, but they can be appreciated from retrospective studies of individual centres and also from registry studies [37,38].

However, the view that MP, in fact, enhances the kidney graft’s tolerance against a prolonged cold ischaemia has yet to be tested against evidence-based criteria. So far, it has only been based on small, monocentric, and retrospective observational studies [39,40,41,42]. In a post hoc analysis, the investigators of the European multicentre study examined the effect of MP in the strata of cold ischaemia. They found that cold ischaemia time remained an independent risk factor for DGF even in machine-perfused kidneys, both in kidneys from brain-dead donors, including ECDs, and in donors after circulatory death [43]. These data barely support that hypothermic MP is able to significantly protect the renal graft from injury caused by prolonged cold ischaemia.

Basically, it is well conceivable that the idea of an increased ischaemia tolerance in machine-perfused kidneys meant that transplants were performed under less time pressure. The acceptance of a somewhat prolonged cold ischaemia at the transplant centre level also allows a longer dialysis time immediately before the operation. The question therefore arises as to whether the recipients of a machine-perfused kidney were dialysed for longer before the operation than control subjects. A consecutively lower serum creatinine and BUN may have led to a postponement of dialysis treatment prior to the onset of graft function, thereby reducing its incidence in the MP group. In such a scenario, the more frequent use of post-operative dialysis in the controls is not necessarily an indication of impaired early graft function after transplantation, particularly when only a singular dialysis is performed. If there was indeed a statistical correlation between dialysis time before and dialysis incidence after surgery, it should be analysed whether the effect of MP on a reduced DGF rate is maintained with a comparable preoperative dialysis time. Additional sensitivity analyses using the need for >1 post-operative dialysis as an outcome measure, including the duration of DGF, could be helpful to better detect graft dysfunction. Data on renal function, particularly calculations of eGFR, e.g., on post-operative day 7, could enable an objective assessment of whether MP has actually improved early function after kidney transplantation. The proposed post hoc analyses are feasible with reasonable effort, as the required data are routinely collected from each transplant recipient. They could demonstrate the efficacy of MP beyond the reduction in an ambiguous treatment-related endpoint that is susceptible to indication bias.

## 6. Summary

All of the procedures discussed in this review article have reduced the dialysis frequency after kidney transplantation in large multicentre studies. The clinical benefit for a broad applicability of therapeutic donor hypothermia has been somewhat tempered by recent controlled data, as it appears to be effective in reducing the incidence of dialysis only in kidneys from ECDs [6,15]. It has also not been shown that the method improves graft function. The advantage is that therapeutic hypothermia can be carried out easily and without side effects in brain-dead organ donors in the ICU before organ removal and does not incur any additional costs.

Donor dopamine, as an example of antioxidant organ conditioning prior to transplantation, led dose-dependently with the infusion time to improved renal function and greater freedom from dialysis during the first week after transplantation in a randomised clinical trial [7]. When analysed in terms of the intention to treat, donor dopamine had no significant effect on graft survival, but was associated with a survival benefit after 5 years if the dopamine infusion was administered for >7 h until cross-clamp [27]. Based on experimental and clinical data, dopamine’s protection against cold ischaemia injury is not limited to the kidneys [44,45]. Dopamine also improved the survival of heart transplant recipients from multi-organ donors included in the dopamine trial [46]. Dopamine was largely eliminated from intensive-care therapy due to its ineffectiveness in stabilising renal function in the critically ill. Due to the negative press of the substance in intensive-care medicine, it is not very likely that the promising approach of donor conditioning with dopamine will be pursued further clinically or experimentally in order to clarify unanswered questions.

In contrast, hypothermic pulsatile MP, in particular, has attracted worldwide interest. At the same time, the available data have been accepted somewhat prematurely, which has already led to the widespread use of the method in clinical practice. In the meantime, it has become clear that MP has no effect on graft survival after kidney transplantation. It also remains unclear whether MP actually improves early renal function. The European multicentre study was unable to demonstrate any effect on renal function 14 days after transplantation [8], and more recent controlled data have not yet convincingly demonstrated this either [9]. This is due, in part, to the ambivalence of the chosen primary study endpoint, which corresponds to the internationally used definition of DGF, but is extremely susceptible to external influences and, therefore, not specific to an assessment of early graft function. Additional sensitivity analyses, which could easily be carried out at low expense, could provide evidence-based information on whether the considerable logistical and cost effort associated with MP really justifies the routine use of the procedure.

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
