# Peer review of "Donor Conditioning and Organ Pre-Treatment Prior to Kidney Transplantation: Reappraisal of the Available Clinical Evidence"

_jcm, 2024, doi:10.3390/jcm13144073_

Round 1
Reviewer 1 Report
Comments and Suggestions for Authors
It is with great interest and pleasure that I read this well-written review on the therapeutic measures aimed at optimizing donor renal function before transplantation. One of the most important things for nephrologists is getting a well-prepared kidney graft for their patients, and, if possible avoiding delayed graft function (DGF). This paper systematically summarizes data on techniques that could help reduce the rate of DGF and is a good source for nephrologists to get insight into available options.
Minor remarks:
- It is suggested to avoid personal phrases and language (“we”, “our”, etc.) and change it to
general terms (“this study”, “close author group”, etc.)
Comments on the Quality of English LanguageIt is with great interest and pleasure that I read this well-written review on the therapeutic measures aimed at optimizing donor renal function before transplantation. One of the most important things for nephrologists is getting a well-prepared kidney graft for their patients, and, if possible avoiding delayed graft function (DGF). This paper systematically summarizes data on techniques that could help reduce the rate of DGF and is a good source for nephrologists to get insight into available options.
Minor remarks:
- It is suggested to avoid personal phrases and language (“we”, “our”, etc.) and change it to
general terms (“this study”, “close author group”, etc.)
Author Response
Reviewer 1
It is with great interest and pleasure that I read this well-written review on the therapeutic measures aimed at optimizing donor renal function before transplantation. One of the most important things for nephrologists is getting a well-prepared kidney graft for their patients, and, if possible avoiding delayed graft function (DGF). This paper systematically summarizes data on techniques that could help reduce the rate of DGF and is a good source for nephrologists to get insight into available options.
Minor remarks:
- It is suggested to avoid personal phrases and language (“we”, “our”, etc.) and change it to general terms (“this study”, “close author group”, etc.)
Response:
Thank you very much for your positive comment. Following your suggestion, we have reworded the two sentences in "Therapeutic donor hypothermia" in lines 87-89 accordingly in order to avoid personal phases and formulations.
Reviewer 2 Report
Comments and Suggestions for Authors
Congratulation to authors, nice, comprehensive and detailed review.
Limited effect of MP on graft function seems to disappoint the authors. However, authors do not appropriately mention and discuss significant prolongation of conservation time, except for rows 265-267 where moderate extension of cold ischemia time was mentioned. MP allows significantly increase storage time which simplify logistics and improves HLA match. Simpler logistics are not limited only to avoidance of nocturnal transplants, mentioned in rows 279-271. More important the allowance to improve HLA compatibility. Better HLA match influences long term survival behind 3-year observation, these long follow up times are not realistic in prospective RCT, they can be appreciated not only in single center retrospective trials, but in registry trials as well. Some reflections on this issue would improve the paper performance.
Author Response
Reviewer 2
Congratulation to authors, nice, comprehensive and detailed review.
Limited effect of MP on graft function seems to disappoint the authors. However, authors do not appropriately mention and discuss significant prolongation of conservation time, except for rows 265-267 where moderate extension of cold ischemia time was mentioned. MP allows significantly increase storage time which simplify logistics and improves HLA match. Simpler logistics are not limited only to avoidance of nocturnal transplants, mentioned in rows 279-271. More important the allowance to improve HLA compatibility. Better HLA match influences long term survival behind 3-year observation, these long follow up times are not realistic in prospective RCT, they can be appreciated not only in single center retrospective trials, but in registry trials as well. Some reflections on this issue would improve the paper performance.
Response:
We would like to thank you for your comprehensive comment, which we fully agree with. We have expanded the discussion on the important issue, that the potential to prolong cold ischaemia in MP kidneys would also facilitate an optimal HLA match between donors and recipients. According to your suggestion we have implemented the following sequence in lines 269-273. “…Apart from simpler logistics in the transplant centres, allowing longer cold ischaemia times would also facilitate improving HLA compatibility. A better HLA match influences long-term survival beyond 3 years of observation (35,36). Longer follow-up times are not realistic in prospective RCTs, but can be appreciated from retrospective studies of individual centres and also from registry studies (37,38)…” We also added references 35-39 to the bibliography and have renumbered it accordingly. However, the available evidence at the molecular level, as briefly summarised under "Dopamine" and also the stratified analysis by Kox et al. (40) does not support the idea that hypothermic MP is able to prevent cold perfusion injury, which would allow a significant prolongation of cold ischaemia time.
Reviewer 3 Report
Comments and Suggestions for Authors
In this review article the authors re-evaluated the available clinical evidence on the basis of the large multicentre controlled trials in order to assess the significance of some therapeutic measures for use in routine clinical practice prior to kidney transplantation. The manuscript is interesting and current, considering the high incidence of important risk factors to kidney function. For example, we can consider how many people suffer from diabetes. Therefore, the authors have the merit to investigate the utility of some interventions in order to improve the success rate of kidney transplantation. The paper is well written and well presented.
Comments on the Quality of English LanguageMinor language editing is required
Author Response
Reviewer 3
In this review article the authors re-evaluated the available clinical evidence on the basis of the large multicentre controlled trials in order to assess the significance of some therapeutic measures for use in routine clinical practice prior to kidney transplantation. The manuscript is interesting and current, considering the high incidence of important risk factors to kidney function. For example, we can consider how many people suffer from diabetes. Therefore, the authors have the merit to investigate the utility of some interventions in order to improve the success rate of kidney transplantation. The paper is well written and well presented
Minor language editing is required
Response:
Thank you very much for reviewing the manuscript and your nice comment. According to your proposal we have made some linguistic changes which are highlighted in the text of the revised manuscript.